TECHNICAL RELEASE

# Snakemake workflows for long-read bacterial genome assembly and evaluation

Peter Menzel[1],*

**1** Labor Berlin - Charité Vivantes GmbH, Sylter Str. 2, 13353, Berlin, Germany

## ABSTRACT

With the advancement of long-read sequencing technologies and their increasing use for bacterial genomics, several methods for generating genome assemblies from error-prone long reads have been developed. These are complemented by various tools for assembly polishing using either long reads, short reads, or reference genomes. End users are therefore left with a plethora of possible combinations of programs for obtaining a final trusted assembly. Hence, there is also a need to measure the completeness and accuracy of such assemblies, for which, again, several evaluation methods implemented in various programs are available. In order to automatically run multiple genome assembly and evaluation programs at once, I developed two workflows for the workflow management system Snakemake, which provide end users with an easy-to-run solution for testing various genome assemblies from their sequencing data. Both workflows use the conda packaging system, so there is no need for manual installation of each program.

**Availability & Implementation:** The workflows are available as open source software under the MIT license at github.com/pmenzel/ont-assembly-snake and github.com/pmenzel/score-assemblies.

**Subjects** Software and Workflows, Bioinformatics, Molecular Genetics

**Submitted:** 11 August 2023

\* E-mail: pmenzel@gmail.com

Preprint submitted at https://doi.org/10.20944/preprints202208.0191.v1

## INTRODUCTION

In recent years, long-read sequencing data from Oxford Nanopore (ONT) sequencers has been widely available for the assembly of microbial genomes. However, *de novo* genome assemblies from ONT sequencing reads contain systematic errors introduced by the comparatively high error rate in the sequencing reads. The major error types are insertions and deletions ("indels") in nucleotide homopolymers. These occur if bases are either skipped or read multiple times during sequencing due to varying translocation speeds [1, 2]. Indels often cause frameshifts in translated protein-coding sequences, leading to issues with *de novo* gene prediction and genome annotation [3]. Such assembly errors can be corrected using the original long reads for "polishing" the primary *de novo* assembly in a consensus step. Further, additional short-read sequencing data, e.g., from Illumina, can also be used to correct assembly errors, either by using both long and short reads together in a "hybrid assembly" [4], or by polishing the long-read assembly using the short reads. If other genome assemblies for the particular organism or assemblies from phylogenetically closely related bacterial strains or species are already available, assembly errors can also be corrected by using such reference nucleotide or amino acid sequences. In recent years, multiple programs were developed for long-read assembly and polishing, resulting in increasingly complex options for end users on how to arrive at a final trusted assembly.

This leads to the need to evaluate the generated assemblies for their correctness. Again, several methods exist for measuring assembly quality by various metrics, often by comparison to already available genome assemblies from the same or closely related species [5, 6].

The workflow manager Snakemake [7] is based on so-called Snakefiles, in which the user defines rules with instructions on how to run programs for creating output files, often via intermediate steps, from a given set of input files. While this approach is similar to the build system GNU Make, Snakemake offers additional syntax features as well as support for executing the programs on job processing systems. Snakemake also integrates the package management system conda (RRID:SCR_018317; conda.io), so that individual software environments can be defined on a per-rule basis, allowing the automatic installation of the required programs upon executing a Snakemake

workflow. Many bioinformatics programs are already included in the "bioconda"
channel [8].

Here, I describe two easy-to-use Snakemake workflows: **ont-assembly-snake** for generating *de novo* bacterial genome assemblies using long (from ONT) and, optionally, short sequencing reads, and **score-assemblies** for evaluating assemblies by various metrics and generating a summary report for an easy comparison of the measurements.

## IMPLEMENTATION

When Snakemake parses the rules defined in a Snakefile, it creates a directed acyclic graph (DAG). Each rule (edges in the DAG) defines the code for generating a set of output files from a set of input files (vertices). Usually, the creation of the DAG starts with the inputs for a special "all" rule, comprising the list of desired output files for the whole workflow, followed by creating intermediary vertices, which eventually are connected to the input files containing the sequencing data. This list of final output files is typically created by the workflow itself, for example, from a user-provided list of sample names or a list of sequencing data files. However, Snakemake's creation and traversal of the DAG for executing rules allow for an easy and very flexible formulation of the chain of execution of rules and the desired final output files by the user himself, without the need for changing the workflow code. This ability is partially in contrast to the "feed-forward" workflow managers, in which the same defined workflow is executed on each input dataset.

The two workflows presented here consist of a Snakefile containing the workflow logic, as well as accessory files for defining conda environments and scripts for report generation using R Markdown [9].

### ont-assembly-snake

The following programs are included in workflow version 1.4:

| Read filtering | | |
|---|---|---|
| Filtlong | github.com/rrwick/Filtlong | |
| Rasusa | github.com/mbhall88/rasusa | [10] |
| **Long read assembly** | | |
| Raven | github.com/lbcb-sci/raven | [11] |
| Flye | github.com/fenderglass/Flye | [12] |
| Miniasm | github.com/lh3/miniasm | [13] |
| Canu | github.com/marbl/canu | [14] |
| **Hybrid assembly** | | |
| Unicycler | github.com/rrwick/Unicycler | [15] |
| **Long read polishing** | | |
| Racon | github.com/lbcb-sci/racon | [16] |
| Medaka | github.com/nanoporetech/medaka | |
| **Short read polishing** | | |
| Pilon | github.com/broadinstitute/pilon | [17] |
| Polypolish | github.com/rrwick/Polypolish | [18] |
| POLCA | github.com/alekseyzimin/masurca#polca | [19] |
| **Reference-based polishing** | | |
| Homopolish | github.com/ythuang0522/homopolish | [20] |
| Proovframe | github.com/thackl/proovframe | [21] |

This workflow makes use of the above-mentioned possibility to let the user define the chain of rule execution by simply creating empty folders with names containing keywords that describe the desired read filtering, assembly and polishing steps. These folder names are used as input for the "all" rule of the Snakefile. When processing the DAG, Snakemake follows the paths from outputs to inputs and executes the rules for each step, creating intermediate assemblies up to the final assemblies defined by the user. The keywords for some programs can be extended by appending numbers, e.g., for setting the number of iterations for assembly polishing.

These three examples show the chaining of keywords with the "+" character to declare the assembly and polishing steps:

```
          ─── Example folder names ───
1  sample1+filtlongMB500_flye+racon2+medaka
2  sample2_flye4+medaka+polypolish+proovframe
3  sample3+filtlongPC90_raven2+medaka+pilon
```

The first assembly is done by reducing the ONT sequencing reads to contain only 500 megabases from the longest reads with the highest quality scores using Filtlong. Those reads are then assembled with Flye (RRID:SCR_017016), followed by two rounds of polishing with Racon (RRID:SCR_017642), followed by long-read polishing with Medaka (RRID:SCR_018157). The second assembly uses all sequencing reads, which are assembled by Flye, including four rounds of internal polishing, followed by Medaka, short read polishing with Polypolish, and lastly, polishing with Proovframe, which requires at least one provided reference proteome. The third assembly uses only the sequencing reads comprising the top 90% bases by their quality, which are assembled using Raven

(RRID:SCR_016190) with two rounds of internal polishing, followed by long and short read polishing with Medaka and Pilon (RRID:SCR_014731), respectively.

In principle it is very easy to include additional programs in the workflow by simply adding additional rules to the Snakefile.

## score-assemblies

This is a classical workflow that runs the same set of programs on the user-provided assembly files in FASTA format, for example, the assemblies generated by ont-assembly-snake.

The following programs are included in version 1.2:

| | | |
|---|---|---|
| Quast | quast.sourceforge.net/quast | [22] |
| BUSCO | busco.ezlab.org | [23] |
| DNAdiff (MUMmer) | github.com/mummer4/mummer | [24] |
| NucDiff | github.com/uio-cels/NucDiff | [25] |
| Pomoxis | github.com/nanoporetech/pomoxis | |
| IDEEL, using: | github.com/mw55309/ideel | |
| Prodigal | github.com/hyattpd/Prodigal | [26] |
| Diamond | github.com/bbuchfink/diamond | [27] |
| Bakta | github.com/oschwengers/bakta | [28] |

All programs, except BUSCO (RRID:SCR_015008) and IDEEL, require one or more reference assemblies to be provided, with which the input assemblies are compared. Quast (RRID:SCR_001228) provides a summary of the usual assembly metrics, such as the number of contigs or N50, but also aligns the assembly to a reference genome, giving statistics about genome rearrangements, mismatches and indels. Similarly, DNAdiff and NucDiff also report alignment metrics. The program assess_assembly from the Pomoxis package calculates an overall Q-score and error rate, whereas assess_homopolymers calculates the fraction of correct homopolymers for each homopolymer length. BUSCO measures the completeness of single-copy clade-specific core genes in the assemblies. By default, it uses a set of core genes that are near universal in all bacteria, but it is also possible to provide a taxon name for selecting a clade-specific dataset. The IDEEL module runs Prodigal for predicting open reading frames (ORFs), which are then searched against the UniProt [29] "Sprot" database and, optionally, a user-provided set of reference protein sequences. It then collects statistics about the number and length of found alignments to database sequences, which give an idea about the completeness and accuracy of the predicted ORFs.

While some of the included programs perform similar tasks, such as approximating a whole genome alignment, and could therefore be considered redundant, they differ in the used heuristics and reported metrics. Fortunately, the execution time of all programs is quite fast on bacterial genomes, which are only a few megabases long, so that it is computationally not a problem to include them all.

The output of the workflow consists of several summary tables and plots, as well as an overall report in HTML format. The report contains sortable tables with the reported metrics from each program as well as a summary table, which contains all metrics per assembly in one line, allowing an easy comparison of the assemblies.

Optionally, the workflow can also run Bakta [28] for the comprehensive annotation of protein-coding and non-coding genes in each genome, which in turn uses various programs and databases for gene prediction and homology search. This annotation can also be used for submitting assemblies to genome databases.

## AVAILABILITY OF SOURCE CODE AND REQUIREMENTS

- Project name: ont-assembly-snake, score-assemblies
- Project home page: github.com/pmenzel/ont-assembly-snake, github.com/pmenzel/score-assemblies
- Operating system(s): Linux
- Programming language: Snakemake
- Other requirements: conda with channels conda-forge and bioconda
- License: MIT
- RRID: SCR_024898, SCR_024897
- WorkflowHub: 10.48546/workflowhub.workflow.786.1, 10.48546/workflowhub.workflow.787.1.

## DATA AVAILABILITY

A small test dataset with ONT and Illumina reads from a *Pandoraea commovens* isolate of a nosocomial outbreak [30] is available on GitHub [31]. The two workflows are archived in workflowhub [32, 33], and snapshots of the code and test data are in GigaDB [34].

## ABBREVIATIONS

DAG: directed acyclic graph; ONT: Oxford Nanopore Technologies; ORF: Open Reading Frame.

## DECLARATIONS

### Ethics approval and consent to participate

Not applicable.

### Competing interests

The author declares that they have no competing interests.

### Author's contributions

PM implemented the workflows described in this manuscript and wrote the manuscript.

### Funding

Not applicable.

### Acknowledgements

Not applicable.

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
