## [Reviewer Report]

Comments on revised manuscriptI'm happy with the changes made now. The example data runs through perfectly fine.  There is a warning printed to the screen when running the assembly workflow: 
```
/mnt/shared/scratch/tadams/review/ont-assembly-snake-testdata/ont-assembly-snake/Snakefile:86: SyntaxWarning: invalid escape sequence '\.' quit(
```  I've seen this in some of my workflows, in my experience the line number isn't necessarily correct, but it should be easily patched by escaping properly, this will lead to an error with later versions of Snakemake, but I think the manuscript should be published and a small patch applied by the author to avoid this warning.

---

## [Editor Report]

Editor’s AssessmentThis work describes and presents two new workflows for the workflow management system Snakemake for bacterial genome assembly and evaluation of assemblies. With the rise of long-read sequencing technologies and their more widespread use for bacterial genomics, there are a number of methods and associated tool for generating genome assemblies from error-prone long reads have been developed. And plethora of possible combinations of programs for obtaining a final trusted assembly. Hence there is the need for measuring completeness and accuracy of such assemblies, for which several evaluation methods implemented in various programs are available. After testing the authors needed to polish the end product and also provide more documentation. These workflows now provide end users with an easy-to-run method for both task and are available as open source software under the MIT license.

---

## [Reviewer Report]

Comments on revised manuscriptThe current version sligthly improves the previous version of the pipelines and incorporates some of the changes that were suggested in the revision, such as the inclusion of a small test set. The current version could be accepted.

---

## [Reviewer Report]

Comments on revised manuscriptWe are grateful for the author adding an example dataset, however when we attempt to run through this, we find an error occurs that causes the workflow to exit. The below output is what is shown when the instructions on the example repository are followed: 
```
$ snakemake -s ont-assembly-snake/Snakefile --use-conda --cores 10 --configfile samples.yaml --config genome_size=5.9 --config medaka_model=r941_min_sup_g507 filtlong min. read length = 1000 Medaka model = r941_min_sup_g507 SystemExit in file /mnt/shared/scratch/tadams/paper_review/ont-assembly-snake-testdata/ont-assembly-snake/Snakefile, line 102: Error: must provide target genome size when using Canu assembler, use option (e.g. for 5.2Mb): --config genome_size=5.2 File "/mnt/shared/scratch/tadams/paper_review/ont-assembly-snake-testdata/ont-assembly-snake/Snakefile", line 102, in <module> File "<frozen _sitebuiltins>", line 26, in __call__
```  This looks to be either an issue with config parsing or an issue with the instructions on the repository.  In terms of the other elements on the release checklist that were not answered with a Yes before: - Guidelines on how to contribute: These are not present.  - Is the code executable: The code fails during a test run on our system.  - Concerns about Snakemake version: This is more flexible now, however it is set to Snakemake versions below 8. I understand Snakemake > 8 does require changes to the codebase, however it's my understanding that Snakemake < 8 will not be supported going forward, so the author may wish to reconsider this choice to provide long-term functionality.  - Information on seeking assistance: There are no details on the repository about this.  - Core functionality documentation: This is improved now.  - Comparison to other software: No comparison is made, though as stated before we don't believe this is critical.  - Test data: This is now present  - Examples: There is an example described in the manuscript within the data availability statement  In terms of content of the manuscript, we do still feel there is too much emphasis on the error rates of long reads, given the latest chemistry has now been out for several months, there is ample evidence that polishing is no longer necessary for these data due to an improved error rate, and that polishing can in fact introduce errors. However, we do accept polishing may be necessary for data generated with legacy chemistry.  I would want to see the errors in the example workflow fixed prior to publication, though the manuscript and documentation are now much improved.

---

## [Reviewer Report]

Reviewer name and names of any other individual's who aided in reviewerJessica Gomez-GarridoDo you understand and agree to our policy of having open and named reviews, and having your review included with the published manuscript. (If no, please inform the editor that you cannot review this manuscript.)YesIs the language of sufficient quality?YesPlease add additional comments on language quality to clarify if neededIs there a clear statement of need explaining what problems the software is designed to solve and who the target audience is? YesAdditional CommentsThe author clearly states in the introduction that his pipelines aim to put several programs for assembly and evaluation of ONT-based bacterial genomes together into a single workflow in order to ease end user's life. Is the source code available, and has an appropriate Open Source Initiative license <a href="https://opensource.org/licenses" target="_blank">(https://opensource.org/licenses)</a> been assigned to the code?YesAdditional CommentsAll the code is available in GitHub under the MIT licenseAs Open Source Software are there guidelines on how to contribute, report issues or seek support on the code?YesAdditional CommentsAlthough it is not explicitly specified, since it is available in a GitHub repository, people can create issues or pull requests to it. Is the code executable?YesAdditional CommentsIs installation/deployment sufficiently outlined in the paper and documentation, and does it proceed as outlined?YesAdditional CommentsIs the documentation provided clear and user friendly?YesAdditional CommentsThe documentation explains well the steps that have to be taken in order to run the pipelines. Is there enough clear information in the documentation to install, run and test this tool, including information on where to seek help if required?YesAdditional CommentsIs there a clearly-stated list of dependencies, and is the core functionality of the software documented to a satisfactory level?YesAdditional CommentsYes, the only dependency is conda and all the needed packages are directly installed by itHave any claims of performance been sufficiently tested and compared to other commonly-used packages? NoAdditional CommentsThere are a few pipelines that also combine several assemblers, advantages or disadvatages of this one in comparison with the others, could be added. However, this is not strongly required, since in the end it is running third-party programs. Is test data available, either included with the submission or openly available via cited third party sources (e.g. accession numbers, data DOIs)?NoAdditional CommentsI have not found any test data in the software distribution. I do recommend that the user includes a couple of small fastq files that could be used to test the installation of the different programs . I had to use one of my own datasets to run the tests. Are there (ideally real world) examples demonstrating use of the software? NoAdditional CommentsThe paper describes the steps that need to be taken in order to run the pipelines, but it could include a section describing the results of at least one run with real data. The way it is written now, it ressembles just a documentation section. Is automated testing used or are there manual steps described so that the functionality of the software can be verified?NoAdditional CommentsAll steps seem to be properly automated, but since these are two different pipelines, there is a manual step in between them. This could be avoided if both workflows were included into a single modular pipeline (See Snakemake documentation https://snakemake.readthedocs.io/en/stable/snakefiles/modularization.html.) Moreover, the way the user specifies the programs that need to be run is by manually creating directories in the expected way, I would recommend using a configuration file instead (See https://snakemake.readthedocs.io/en/stable/snakefiles/configuration.html). This would make it more intuitive for the user and get rid of the manual steps. Any Additional Overall Comments to the AuthorThese pipelines can be of great use for the community, so I recommend its publication. However, I do think it is necessary to at least include a test dataset and an additional section in the paper describing a complete run with real data. Moreover, there are a few changes to the pipelines that would greatly improve its functionality. Snakemake allows the integration of different modules into the same pipeline, so I'd strongly recommend to join the two workflows into a single one and give it the ability to run either independently or together. This would make it possible to evaluate the obtained assemblies in the same run, making the users life easier and totally eliminating manual steps. Additionally, you could get the stats of some of the assemblies while maybe other programs are still running.  On another hand, the author gives flexibility to its pipelines by having the user create directories with a certain nomenclature so Snakemake knows which programs to run. Although this is not a bad idea, it can get complicated if the user wants to run all the available programs. I would strongly recommend to incorporate a config file to the pipeline where the user can specify exactly the programs and parameters he/she wants to run. RecommendationMajor Revisions

---

## [Reviewer Report]

Reviewer name and names of any other individual's who aided in reviewerThomas Adams and Moray SmithDo you understand and agree to our policy of having open and named reviews, and having your review included with the published manuscript. (If no, please inform the editor that you cannot review this manuscript.)YesIs the language of sufficient quality?YesPlease add additional comments on language quality to clarify if neededLanguage is clear throughoutIs there a clear statement of need explaining what problems the software is designed to solve and who the target audience is? YesAdditional CommentsIt would help if the author further clarified why their solution is the best choiceIs the source code available, and has an appropriate Open Source Initiative license <a href="https://opensource.org/licenses" target="_blank">(https://opensource.org/licenses)</a> been assigned to the code?YesAdditional CommentsAn MIT license is usedAs Open Source Software are there guidelines on how to contribute, report issues or seek support on the code?NoAdditional CommentsThe GitHub repository allows issues and pull requests, but there is no clear guide on preferred formats for issues or how to format contributions via pull requests.Is the code executable?Unable to testAdditional CommentsThere is no example data to allow proper testing of the workflow. This would be extremely beneficial as it would allow bug reports to confirm whether their pipeline is correctly set up.Is installation/deployment sufficiently outlined in the paper and documentation, and does it proceed as outlined?YesAdditional CommentsInstallation consists of cloning a github repository and provisioning an initial conda environment. I'm not clear why the yaml explicitly requests snakemake 7.24, this is now 8 months old, would it not be better to use the latest release? If there is a reason it should be explained. Our conda setup struggled to create the environment for assessing assemblies, I expect as we have a higher version of Snakemake installed in base.Is the documentation provided clear and user friendly?YesAdditional CommentsThe documentation is clear.Is there enough clear information in the documentation to install, run and test this tool, including information on where to seek help if required?NoAdditional CommentsThere's no information on how best to seek assistance if issues arise, but the install instructions are well written.Is there a clearly-stated list of dependencies, and is the core functionality of the software documented to a satisfactory level?NoAdditional CommentsWe am unable to assess functionality without test data.Have any claims of performance been sufficiently tested and compared to other commonly-used packages? Not applicableAdditional CommentsThere are no claims comparing it to other software. This would strengthen the manuscript but is not critical.Is test data available, either included with the submission or openly available via cited third party sources (e.g. accession numbers, data DOIs)?NoAdditional CommentsThere is no test dataAre there (ideally real world) examples demonstrating use of the software? NoAdditional CommentsThere is an example output description for score-assemblies, but not for ont-assembly-snakemakeIs automated testing used or are there manual steps described so that the functionality of the software can be verified?YesAdditional CommentsThere are some lints used, though these look to validate format rather than functionality of the workflows.Any Additional Overall Comments to the AuthorThese workflows could well become a useful tool for researchers. It's definitely important to encourage researchers to use multiple methods of assembly and evaluate them.  The workflows themselves could use some polishing. For instance, the required directory structure could be replaced with a config yaml file, meaning users aren't spending time copying files into new locations and can instead just declare paths. We would prefer to avoid being strict on file extensions too, though if these are a limitation of the tools used in the workflow that can't be helped.  Within the rules themselves, the author may wish to consider greater use of the temp() function to reduce storage space used. It would also be helpful to have memory requirement specified within each rules as a resources flag. I'd imagine most users of this software will be on compute clusters, where it is usually a requirement to specify a memory cap in advance and we have found snakemakes own estimates of memory usage to be extremely poor.  Finally, the user discusses a lot about inaccuracies in long reads and the need to polish, it is our understanding that whilst this was definitely the case with early ONT reads, the latest chemistry now reaches over 99% accuracy (Q30) and I've seen many discussions amongst bioinformations suggesting that the risk of introducing additional errors from polishing renders them obsolete. These improvements are discussed in this paper: https://www.nature.com/articles/s41592-022-01716-8 we would also exercise caution with hybrid assemblers, in our experience these have never performed brilliantly and with the improvements in long read sequencing it's difficult to see how introducing inherently hard to assemble short reads would provide a benefit.  In terms of assessment, we would suggest the author consider integrating tools like merqury, which use kmer spectra of short reads to assess your assemblies and in our hands has performed well. https://github.com/marbl/merqury
RecommendationMajor Revisions

---

## [Reviewer Report]

Reviewer name and names of any other individual's who aided in reviewerZhigui BaoDo you understand and agree to our policy of having open and named reviews, and having your review included with the published manuscript. (If no, please inform the editor that you cannot review this manuscript.)YesIs the language of sufficient quality?YesPlease add additional comments on language quality to clarify if neededIs there a clear statement of need explaining what problems the software is designed to solve and who the target audience is? YesAdditional CommentsIs the source code available, and has an appropriate Open Source Initiative license <a href="https://opensource.org/licenses" target="_blank">(https://opensource.org/licenses)</a> been assigned to the code?YesAdditional CommentsAs Open Source Software are there guidelines on how to contribute, report issues or seek support on the code?YesAdditional CommentsIs the code executable?YesAdditional CommentsIs installation/deployment sufficiently outlined in the paper and documentation, and does it proceed as outlined?YesAdditional CommentsIs the documentation provided clear and user friendly?YesAdditional CommentsIs there enough clear information in the documentation to install, run and test this tool, including information on where to seek help if required?YesAdditional CommentsIs there a clearly-stated list of dependencies, and is the core functionality of the software documented to a satisfactory level?YesAdditional CommentsHave any claims of performance been sufficiently tested and compared to other commonly-used packages? YesAdditional CommentsIs test data available, either included with the submission or openly available via cited third party sources (e.g. accession numbers, data DOIs)?NoAdditional CommentsAre there (ideally real world) examples demonstrating use of the software? YesAdditional CommentsIs automated testing used or are there manual steps described so that the functionality of the software can be verified?NoAdditional CommentsAny Additional Overall Comments to the AuthorIn reviewing the paper "Snakemake workflows for long-read bacterial genome assembly and evaluation," I find the overall work impressive but suggest a few minor revisions for enhanced clarity and usability. Firstly, the inclusion of Directed Acyclic Graph (DAG) flowcharts for each pipeline would significantly aid users in visualizing and understanding the workflow processes, thereby demystifying the sequence of operations and their dependencies. Secondly, providing small test datasets, similar to the Canu E. coli test data, which users can download from databases like NCBI or ENA, would allow for a more hands-on, user-friendly experience. These test datasets are invaluable for users new to the system, enabling them to familiarize themselves with the workflow using manageable data sizes. Lastly, the addition of a module to visually display the assembly graph would be particularly beneficial for analyzing bacterial genomes, especially to ascertain if they are circular and to identify the presence of circular plasmids in the final assembly. Moreover, incorporating an FAQ section based on common user issues would further streamline the learning process and enhance the overall utility of the workflows. These minor revisions would undoubtedly elevate the paper's practicality and accessibility, making it a more comprehensive guide for users engaging with long-read bacterial genome assembly and evaluation.RecommendationMinor Revisions